# Family-Centered Early Intervention (FCEI) Involving Fathers and Mothers of Children Who Are Deaf or Hard of Hearing: Parental Involvement and Self-Efficacy

**DOI:** 10.3390/jcm11030492

**Published:** 2022-01-19

**Authors:** Evelien Dirks, Amy Szarkowski

**Affiliations:** 1Dutch Foundation for the Deaf and Hard of Hearing Child (NSDSK), 1073 GX Amsterdam, The Netherlands; 2Department of Psychology, Utrecht University, 3508 TC Utrecht, The Netherlands; 3Children’s Center for Communication/Beverly School for the Deaf (CCCBSD), Beverly, MA 01915, USA; Amy.Szarkowski@childrens.harvard.edu; 4LEND (Leadership Education in Neurodevelopmental and Related Disabilities) Program, Division of Developmental Medicine, Boston Children’s Hospital, Boston, MA 02215, USA; 5Department of Psychiatry, Harvard Medical School, Boston, MA 02215, USA

**Keywords:** family-centered early intervention (FCEI), deaf, hard of hearing, deaf or hard of hearing (DHH), fathers, mothers, self-efficacy, involvement, perceived support, early intervention (EI)

## Abstract

(1) Background: Studies related to family-centered early intervention (FCEI) for children who are deaf or hard of hearing (DHH) have largely focused on mothers, at the exclusion of fathers. Yet, understanding fathers’ experiences with FCEI is also important and may inform service delivery. The present study explores self-efficacy and involvement with FCEI in both fathers and mothers. (2) Methods: Dutch fathers and mothers completed questionnaires about their parental self-efficacy, involvement in FCEI, perceived support from their primary EI provider, and the impact of raising a child who is DHH on parenting. (3) Results: Both fathers and mothers reported relatively high levels of self-efficacy. Mothers reported higher levels than fathers on some domains of self-efficacy and tended to be more involved in their child’s FCEI than fathers. In fathers, but not mothers, higher levels of self-efficacy were related to higher levels of involvement and higher levels of perceived support. (4) Conclusions: Similarities and differences were found between fathers and mothers in their perspectives on self-efficacy and involvement. This points to potential differences related to their FCEI needs. EI providers need to address both the needs of fathers and mothers to promote optimal development among child who are DHH.

## 1. Introduction

Family-centered early intervention (FCEI) focuses the delivery of early intervention (EI) supports on the caregivers of children who are deaf or hard of hearing (DHH), which can include extended family members and/or other caregivers in the community surrounding the child [1]. Naturally, children can be raised by biological parents, step-parents, multiple parents, grandparents, and others who have assumed the role of caregivers. The authors acknowledge that family constellations can vary widely. It should not be assumed that a child has “one mother and one father”, although this remains true for many DHH children. Historically, much focus has been centered on mothers of children who are DHH at the exclusion of fathers. FCEI service provision is intended to address the needs of the whole family, yet without the inclusion of fathers (and/or others actively caring for young DHH children) in studies related to FCEI, EI providers, EI programs and systems, researchers will have a less comprehensive understanding of the different ways that fathers and mothers may experience, perceive of, engage with, and learn from the support offered by FCEI.

### 1.1. The Importance of Fathers

It is widely recognized that children develop within the context of their relationships with caregivers and others, and that fathers play an important role [2,3]. Both the amount of time that fathers spend with their child (father involvement) and the quality of their interactions (father–child relationship) are important, yet are often conflated in the literature on child and human development [4]. Examining the relationships between fathers and children can yield valuable information about fathers’ sensitivity toward and responsiveness to the needs of their children. The measurement of fathers’ involvement (whether conceptualized as presence/absence or as engagement) can provide valuable information about fathers’ roles in their children’s lives in general or in specific aspects of their lives, such as involvement in FCEI services.

Studies of father–child relationships and father involvement in their children’s lives collectively suggest that fathers do have an impact; their presence and engagement have been linked to child outcomes, such as development of secure attachments [4,5], children’s emotional regulation [6], the father–child bond/attachment [7] and a number of important social determinants of health [8]. Across studies that have explored the outcomes of fathers of children with chronic illness, better outcomes are associated with higher quality father–child relationships and with more father involvement—both across general activities as well as activities that are related to supporting the child’s disability or condition [9]. A review of fathers’ experiences and perceptions of parenting children with developmental disabilities found that adaptation to the child’s disability, planning for the future, and involvement with health services and supports for the child helped to shape their participation in family life [10]. Fathers of children with intellectual disabilities who employ “mindful approaches” to parenting have been found to be more highly involved in child-related parenting tasks and report lower levels of parental stress, which may be linked to improved parent–child interactions [11]. Fathers themselves also benefit from being connected to and involved with their children [12]. Fathers both actively shape and are shaped by their interactions with their children. There is value in exploring fathers’ experiences with FCEI.

In a literature review on the perceptions and experiences of parents of DHH children of 0–6 years of age, which spanned 50 years of research, just 37 studies were identified that explicitly included fathers [13]. The body of research that includes fathers of young DHH children can generally be classified into six themes: (1) fathers’ perspectives on parenting; (2) perceived stress and coping among fathers; (3) aspects of parent–child interactions; (4) benefits of fathers being included in their children’s lives; (5) fathers’ parental self-efficacy; and (6) fathers’ involvement in early intervention. The latter two topics, fathers’ *involvement* and *parental self-efficacy*, are of particular interest to the present paper.

### 1.2. Parental Self-Efficacy

Parental self-efficacy (PSE), derived from Bandura’s [14] description of self-efficacy in the Social Cognitive Theory, refers to parents’ beliefs about their ability to successfully perform in their parenting role and their belief about their ability to successfully raise children [15]. PSE differs from parental confidence, which is defined as more general and global in nature, and parental self-esteem, which more closely describes parents’ judgements about their worth as a parent. PSE is situation-specific and context-dependent; it reflects parents’ self-judgement about their ability to fulfil the parenting role. It includes an assessment of the strength of the belief and, to some extent, an interpretation of that belief as being within one’s own control [15]. Because self-assessment of self-efficacy is context dependent, parents can rate their confidence and abilities across different tasks in different ways [16]. In general, PSE attempts to capture parents’ beliefs about, and sense of competency toward, parenting their child or children.

PSE is informed by a number of factors: (a) the parent’s personal and psychological resources (including what they have learned through vicarious experiences, their interpretation of how they have performed in the parenting role in the past, how they have attributed their actions, etc.); (b) the characteristics of the child (such as temperament and presence/absence of behavioral challenges); and (c) the environmental context (including sources of support and sources of stress, parents assessment of the situation including resources and constraints, etc.) [17,18,19]. PSE is thought to be positively associated with supportive/engaged and positive parenting behaviors [20], positive affect, parental control, and the general self-efficacy beliefs of the parents. It is believed to be inversely related to negative affect, child behavior problems, and hostile or coercive parenting behaviors [21]. PSE has also been demonstrated to be a buffer against parenting stress and has been linked to better child development outcomes and enhanced child psychosocial adjustment [17].

#### 1.2.1. Parental Self-Efficacy: Differences in Fathers and Mothers

The drivers of PSE in fathers and mothers are similar in some ways. Hypothesizing that child factors, parent factors, and family factors influence parental self-efficacy, Giallo and colleagues [22] conceptualized PSE as a mediator between that collection of factors and parental involvement. Indeed, their research showed that PSE mediated the pathways between both: (1) stress, anxiety, and depression (parent factors); and (2) difficult temperament (a child factor) and the relationship with parental involvement for mothers and fathers. This effect was moderate; parents who rated themselves lower in self-efficacy were less involved with their children. In both fathers and mothers, PSE is informed by a general sense of competence and the extent to which a parent of either gender holds the ‘parenting role’ to be central to their identity [21]. High PSE is associated with positive affect and with increased levels of engaged parenting behaviors for both fathers and mothers.

Although fathers have largely been underrepresented in research regarding parental self-efficacy [21], some research on PSE has found differences between fathers and mothers. Fathers’ relationships with their families, including marital satisfaction and family functioning satisfaction, as well as the extent to which fathers experience parental stress seems to predict PSE; for mothers, on the other hand, general self-efficacy and overall family functioning predict PSE [23]. Maternal PSE, but not paternal PSE, has been shown to be significantly and negatively associated with child behavior problems and parental negative affect [21]. Fathers’ PSE is also associated with parental control; whether they perceive themselves as having effectively “controlled their child’s behavior” does seem to influence fathers’ estimation of abilities more so than for mothers [21].

A systematic review of PSE [15] documented the need to better understand differences in PSE among mothers as well as fathers. Recommendations for studies of PSE in general early intervention have also argued for the explicit inclusion of fathers [16].

#### 1.2.2. Parental Self-Efficacy: Fathers and Mothers of Children Who Are Deaf or Hard of Hearing

Specific to children who are DHH, PSE informs the parental goals of EI; mothers who report higher levels of perceived knowledge and competence in developing their children’s language abilities provide higher-level language strategies (fathers were not included in this particular study) [24]. Parents of children who use cochlear implants report higher levels of PSE than parents whose children utilize hearing aids, perhaps related to the amount of support they receive in understanding the technology and following through with audiological recommendations [25]. No statistical relationship was found in a study examining PSE and involvement with establishing EI goals (as determined through the use of the IFSP, or individualized family service plan, in a study conducted in the U.S.) [26]; however, parents who were more heavily involved with the child’s device use were also more involved in FCEI services.

Parents of children who are DHH have been queried with the SPISE (scale of parental involvement and self-efficacy) [24] and SPISE-R (scale of parental involvement and self-efficacy—revised) [27], which can provide information about parents’ knowledge, beliefs, confidence, and actions that pertain to supporting a child’s auditory access and fostering the child’s language development [27]. Overall, both mothers and fathers report relatively high levels of self-efficacy related to caring for and supporting the audiologic needs and language development of children who are DHH [26,28]. Family stress does seem to negatively impact both PSE related to audiological care and parent involvement, at least among children who utilize spoken language [29]. The exploration of PSE of parents/caregivers of children who are DHH outside of the focus of audiologic understanding and management is an area that is, to date, understudied.

In an examination of PSE among parents of children who are DHH and enrolled in EI, the need to further understand PSE in general, among both fathers and mothers, was emphasized [26]. The few studies of PSE among parents of children who are DHH do emphasize the need to better understand the role of fathers [26,27,30]. Fostering parental self-efficacy for all parents/caregivers should be a goal for FCEI for children who are DHH [1].

### 1.3. Parental Involvement in Early Intervention

Parental involvement, as a construct, can be challenging, because the term is used differently in various contexts. As it pertains to FCEI, involvement might be conceptualized as attendance at FCEI sessions, engaging in the service planning and goal setting for the child, taking on a proportion of the “burden of care,” or rated in terms of “investment in the parenting role,” to name a few possibilities. Consequently, involvement can be measured in a number of ways, ranging from the use of a time-diary (as seen in work by McBride and colleagues [31]), to completion of a parental involvement scale (see Giallo et al. [22] as an example), to documenting a parent’s level of responsibility for the daily care of the child (such as Potter [32]), or having one parent rate or describe the level of involvement and help provided by the other parent (as did Simmerman and colleagues [33]). While each lens on the meaning of involvement may have value, it is important to note that comparisons across the literature on this construct are not straightforward.

Parental involvement with their children can be influenced by a number of factors, including: (1) *child factors*, such as temperament [34]; (2) *parental factors*, such as stress or adaptation of the mother/father role identity [12,34]; and (3) *contextual factors*, such as familial socio-economic standing and the quality of the couple relationship [22], employment status and the flexibility of parents’ schedules [22], as well as the setting in which EI is provided [35,36]. A number of studies have highlighted a negative relationship between a greater number of contextual barriers (e.g., lack of or limited employment and inflexible schedules), and parental involvement with the child [37]. In a study of fathers and mothers, full-time employment for both genders influenced the amount of time spent with children; yet, when researchers have controlled for employment status, the differences noted in mothers’ and fathers’ involvement were minimal [22]. Parental involvement has been found to be positively and significantly associated with parental satisfaction with the support that the family is receiving [38].

#### 1.3.1. Fathers Involvement in Early Intervention

Increasingly in recent years, fathers have been more highly involved within family systems, which often has a positive impact on the family [12,31,33,39]. Unfortunately, however, this increased appreciation of the importance of fathers and the unique role that they can play in the lives of their young children has not necessarily shifted how FCEI support is provided to increase their ability to be involved [12,33,40]. Even in programs that celebrate family-centered approaches, programmatic or system-wide efforts to increase the involvement of fathers and better understand the factors that can support father involvement are not common [40].

In contrast to theoretical models that have argued that parents’ beliefs alone drive their behaviors and level of involvement/engagement, independent of social class, geography, cultural background, skills and knowledge [37], Freeman and colleagues [41] have pointed to fathers’ beliefs as being instrumental, suggesting that beliefs about parenting function as a mediator between family contextual factors and father involvement in EI. The extent to which fathers feel empowered in their role as a parent (to help, guide, or support the child) and the extent to which they believe that their role as a parent can have a positive impact on the child can influence fathers’ level of involvement [12]. Increased father involvement in FCEI may help to reduce mothers’ level of parental stress and may contribute to a more active engagement by the family [39].

Fathers seem to be influenced by contextual factors; as the number of barriers to involvement in FCEI increase, fathers are less likely to engage in physical play, participate in caregiving, or be involved in programming or social activities [41]. Yet, when fathers’ sense of PSE and their ideas about their roles as a father of a child in EI are taken into account, certain contextual barriers no longer predicted father involvement [41]. Fathers show benefits of involvement in FCEI, such as experiencing increased self-confidence, feeling more satisfied with the parenting role and more connected to their child, and being less distressed [40].

McBride and colleagues [42] examined the perceptions of EI providers regarding father involvement and found a gap between EI providers’ reported perceptions and their actions. Three categories of barriers to father involvement in EI were found. Two categories, *lack of presence* (e.g., fathers were working, disengaged, or not involved in the child’s life), and *men work and women care for children* (i.e., fathers’ and mothers’ attitudes toward involvement—informed by societal gender roles, perceptions/beliefs about parental efficacy, division of labor, and fathers’ beliefs about their roles with their children), were conceptualized by the researchers as being fairly “set.” In contrast, the third category, *EI providers’ ability to adapt*, was described as being amenable to change. In general terms, father involvement in EI is impacted by the approach that EI providers use with families, as well as their availability and flexibility in providing services. A large study of EI providers’ assessment of their own competence and confidence found that only 30% of EI providers believed themselves to have the skills needed to work effectively with families [43]. An argument can be made for helping EI providers to gain the skills necessary to feel more confident engaging with all caregivers, and in particular, fathers.

#### 1.3.2. Parental Involvement in Early Intervention in Fathers and Mothers of Children Who Are Deaf or Hard of Hearing

Family involvement is a foundational principle in FCEI [1,44]. The level of involvement of mothers of children who are DHH has been linked to personal characteristics (namely anxiety, curiosity, anger, and motivation) as well as contextual mediating variables (i.e., pessimism and informal support) [45]. Increased family involvement in FCEI has been linked to improvements in children’s language development [46]. Parents have reported that several factors influence their involvement in EI in developing nations in particular, including the extent to which services provision considers the resources available to the family, the extent to which service provision is cohesive, the extent to which the reasons for EI are understandable to the family, and the rapport that is built between family members and the EI provider [47]. While the importance/value of the involvement of fathers in FCEI for children who are DHH has been mentioned in various studies, “father involvement” has not been extensively studied.

### 1.4. Parental Self-Efficacy and Involvement in Family-Centered Early Intervention among Fathers and Mothers of Children Who Are Deaf or Hard of Hearing

The present study is informed by the work of colleagues who have also studied PSE and the involvement of parents of children who are DHH. Hintermair and Sarimski [30] researched the experiences of fathers of DHH infants and toddlers, examining how having a child who is DHH impacted fathers, how often fathers attended meetings with EI providers, how intensely fathers perceived themselves to be involved with EI, and what factors contributed to fathers’ self-efficacy. Because that study focused solely on fathers, it asked questions about the marital relationship and the extent to which fathers perceived the EI providers interacting specifically with their wives. Fathers in that study did not necessarily desire to have their own specialized approaches to engaging in EI; rather, they expressed a desire to be more highly involved in the regular EI sessions with their wives and children.

Zaidman-Zait and colleagues [48] explored the experiences of 30 Israeli–Arab couples, assessing the experience of mothers and fathers of children age 3–7 years. They examined parental involvement, parental stress, parental self-efficacy, and social support. Mothers’ self-efficacy and involvement were not related; in the cultural context in which that study occurred, 73% of mothers were not employed outside the home. Involvement was not related to self-efficacy in mothers; the researchers surmised that involvement in EI is a primary role assigned to mothers. In contrast, PSE and involvement were positively associated for fathers. Additionally, support from family and friends was associated with the increased involvement of fathers. The authors emphasized that the experiences of fathers and mothers should not be considered to be interchangeable.

Ingber and Most [49] also studied parental self-efficacy and involvement, as well as child and family characteristics that can contribute to these, in a study of parents of children 3–7 years. Interestingly, they included mothers’ ratings of father involvement. Fathers who scored higher in PSE were rated by their spouses as more involved. In general, fathers were highly involved; the greater empowerment of fathers was recommended as a means to increase fathers’ confidence and PSE. However, as this study focused on fathers of children in preschool, the results are not directly applicable to FCEI.

These three studies provided relevant information and important insights regarding fathers’ and mothers’ PSE and involvement that informed the present study. The latter two studies explored the experiences and perceptions of parents of preschool children who were beyond the age of FCEI; the former included only fathers. The question remained, how might fathers and mothers of young DHH children in FCEI rate their involvement and self-efficacy (and relatedly, their perceived support by the primary EI provider)? What might the potential differences in fathers’ and mothers’ ratings suggest in terms of needs for support in FCEI?

### 1.5. Aims

In the present study, we examine the similarities and differences between fathers and mothers of young children who are DHH related to parental self-efficacy and involvement. These families were enrolled in FCEI services in the Netherlands, receiving support from a primary EI provider. We pose two questions:What are the differences between fathers and mothers in their levels of involvement, their parental self-efficacy in parenting a child who is DHH, and their perceived support from the EI provider?What are the associations between parental self-efficacy, involvement, and perceived support in both fathers and mothers in FCEI?

## 2. Materials and Methods

### 2.1. Recruitment and Participants

Four organizations providing FCEI support to families and children who are DHH in the Netherlands participated. EI providers working in EI centers sent an email containing a flyer about the study to caregivers; those interested responded by emailing the research team to indicate their desire to participate. Researchers acknowledged receipt of caregivers’ emails and sent a personalized code and link to the online questionnaire. Before the caregivers could begin the online questionnaire, they were required to provide informed consent.

A total of 24 Dutch couples with at least one child with a moderate-to-profound hearing loss, ages 0–48 months (4 years), participated in the current study. Demographic information of the caregivers and children is provided in Table 1. The children were between 9 and 48 months of age and received EI during the study. The families of DHH children with additional challenges/disabilities were included in the study as well, and comprised 33% of the sample.

### 2.2. Instruments and Measures

#### 2.2.1. Parenting Self-Efficacy

To assess the caregiver’s PSE, the *Tool to Measure Parenting Self-Efficacy* (TOPSE) [50] was used. The TOPSE is a multidimensional self-report instrument including 48 statements about parenting based on the theoretical framework of Bandura’s social learning theory [14,51]. The TOPSE consists of eight subscales: emotion and affection, play and enjoyment, empathy and understanding, control, discipline and boundaries, pressures, self-acceptance, and learning and knowledge; each scale consisted of six items. The items are rated on an 11-point Likert scale, ranging from 0 (completely disagree) to 10 (completely agree). Example items include: “I am able to show affection toward my child” and “I am able to have fun with my child.” The scale contains positively and negatively worded items, and the responses are summed to create a total score; the lower the score, the lower the level of PSE.

In the current study, the Dutch version of the TOPSE was used [52]. We added one scale to the original TOPSE to assess parenting self-efficacy concerning communication and hearing with items such as “I am able to promote my child’s language development” and “I am able to help my child wear his/her hearing aid and/or cochlear implant.” The overall scale reliability was 0.74; however three subscales (control, pressure, and learning and knowledge) revealed low reliability and therefore three items that were responsible for the low alpha were excluded. Two of these items concerned behaviors that were not specific to the parent–child interaction: “I can remain calm when facing difficulties” and “I can overcome most problems with a bit of advice.” The third item seemed to be difficult for participants to answer, perhaps related to asking respondents to answer a “double negative” question: “I do not feel a need to compare myself to other parents.” The reliability of the emotion and affection scale was also low, attributable to the item, “I find it hard to cuddle my child.” Because this item does reflect self-efficacy, is relevant to our understanding of parent–child interactions, and was not difficult for parents to determine its meaning or respond to this question, we retained this item. For the analysis, only the total TOPSE score was used. The reliability measures after removing the items are presented in Table 2.

#### 2.2.2. Experience with a DHH Child and Involvement in FCEI

To explore caregivers’ experiences with DHH children, we used the Fathers of Children with Developmental Challenges instrument (FCDC) [53]. In line with adaptations made by Hintermair and Sarimski [30], we reformulated the items and replaced the term “disability” with “deaf or hard of hearing.” We also replaced “fathers” with “caregivers,” since we sought input from both fathers and mothers. The FCDC consists of two subscales: “Impact on Parenting” and “Involvement in Child Intervention.” The first subscale, “Impact on Parenting”, includes 12 queries about the impact of the child being DHH on caregivers’ experiences (e.g., “His/her hearing loss gets in the way of our relationship”; “Having a DHH child is more difficult than I expected”). Caregivers rated their experiences using a Likert scale, ranging from 1 (strongly disagree) to 5 (strongly agree). Higher scores suggest larger impact of the child being DHH on caregivers’ experiences.

The second subscale, “Involvement in Child Intervention” addresses caregivers’ experiences regarding opportunities to be involved in EI for their child (e.g., “Meetings with professionals are arranged at a time that fits with my work schedule”; “During meetings regarding my child’s therapies, it is easy for me to have a say”). In the present study, five items were used instead of the eight items in the original form. We omitted two questions related to attendance at educational meetings; as this studied focused on FCEI, we found these questions to be less pertinent. Additionally, a final question asked fathers about whether comments made during sessions involving both parents were addressed to their wives. In contrast to the study by Hintermair and Sarimski [30], in the present study both fathers and mothers were included, and we did not wish to make this distinction. Items on this “Involvement in Child Intervention” subscale were rated using the same 1–5 Likert scale as was used for the “Impact on Parenting” subscale. Higher scores indicate a greater degree of parental involvement in the child’s intervention.

The FCDC was translated into Dutch according to the guidelines of the International Test Commission [54] using a “translation/back-translation” procedure [55]. The reliability (internal consistency) for both scales in the Ly and Goldberg study [53] was satisfactory (Cronbach’s alpha > 0.85). A reliability check with the data from our study showed acceptable values (see Table 2).

#### 2.2.3. Perceived Support by FCEI

To assess caregivers’ perceived support from EI providers regarding their DHH child, we developed a scale based on an informal scale described by Hintermair and Sarimski [30]. Caregivers answered six items, rating the support they received from their primary EI provider (e.g., “The EI provider supports me in promoting my child’s language development”; “The EI provider supports me in handling reactions from my family”), using the same 1–5 Likert scale described above. Higher scores indicated parents’ perception of a higher level of support received from the EI provider. The internal consistency of the scale was good (see Table 2).

#### 2.2.4. Frequency of Participation in FCEI

Three questions were asked to measure caregivers’ frequency of participation in FCEI: (1) frequency of the parent joining the in-home EI session; (2) frequency of parental contact with the primary EI provider; and (3) frequency of parental attendance at courses/training offered at the EI center.

### 2.3. Data Analysis

Because of the small sample size and the fact that not all variables met the assumptions of parametric testing, non-parametric tests were used. Differences between fathers and mothers were examined by using Mann–Whitney tests. Correlations between the measures were calculated with Spearman’s rho correlations.

## 3. Results

### 3.1. Parental Self-Efficacy

Figure 1 depicts fathers’ and mothers’ mean subscale scores for the TOPSE. Mann–Whitney U tests were used to determine whether PSE differed between fathers and mothers. Fathers reported lower PSE than mothers concerning emotion and affection, *U* = 166.50, *p* = 0.012; empathy and understanding, *U* = 165.00, *p* = 0.011; control, *U* = 178.50, *p* = 0.023; learning and knowledge, *U* = 170.00, *p* = 0.015; and play and enjoyment, *U* = 176.00, *p* = 0.021). On the other subscales of the TOPSE, no differences between fathers and mothers were found.

The total PSE score tended to be lower for fathers than mothers, although responses from both parents indicate good PSE (see Table 3). For fathers and mothers, no relationships were found between PSE and child age or parental educational level. The fathers’ numbers of hours engaged in employment was negatively associated with PSE (*r* = −0.56, *p* < 0.01); no similar associations were found for mothers.

### 3.2. Impact on Parenting

No significant differences were found in the overall mean scores between fathers and mothers in their perception of the impact of having a child who is DHH on their parenting (see Table 3). Item-level analysis, however, shows that a significant number of fathers and mothers experience some struggles with having a child who is DHH in the family (see Table 4). For example, more than 30% of the parents experience challenges in being involved in the child’s therapy and/or dealing with the child’s diagnosis. Neither child’s age nor parents’ educational level nor time spent in employment were associated with parents’ perceptions of the impact that the child being DHH had on their parenting.

### 3.3. Involvement in Intervention

Both fathers and mothers rate themselves as ‘highly involved’ with their child’s EI, with at least 75% of the fathers and 83% of the mothers endorsing “agree/strongly agree” on individual questions regarding their involvement. The parental educational level did not influence these results. While the number of hours working was negatively associated with the level of involvement for fathers (*r* = −0.54, *p* < 0.01), this relationship was not noted for mothers. Instead, for mothers, child age was related to involvement (*r* = 0.41, *p* = 0.048); mothers endorsed greater involvement when their child was older, whereas for fathers, the child’s age did not affect their perceived involvement.

### 3.4. Parents’ Frequency of Participation in Early Intervention

Parents were asked about the frequency of their participation in FCEI. Overall, the majority of fathers and mothers do participate in the in-home EI sessions, including 66.6% of fathers and 95.7% of mothers. Fathers do, however, attended at-home intervention sessions less frequently than mothers (*U* = 0.86, *p* < 0.001; Fathers M = 2.58, SD = 1.14; Mothers M = 3.83, SD = 0.64; range 1–4). A total of 33.3% of the fathers never (4.1%) or rarely (29.2%) attended, while 66.6% of fathers attended often (45.8%) or always (20.8%). In contrast, 95.7% of mothers always (91.7%) or often (4.1%) attended in-home EI sessions, while just 4.1% rarely participated. The age of the children was not related to the attendance of house visits, *r* = −0.06, *p* = 0.686. Once again, the fathers’ number of hours of employment per week was negatively related to their attendance, *r* = −0.74, *p* < 0.001, whereas the same relationship between work and attendance in EI was not found for mothers, *r* = −0.33, *p* < 0.001.

We asked the parents about their frequency of contact with their primary EI provider. A majority of the fathers (62.5%) reported having contact less than once a month, 20.8% had contact once a month and 16.7% had contact with their EI provider every two weeks. The percentage of the mothers who had contact with the primary EI provider less than once per month was 41.7%, 50% had contact once a month, and 4.1% had contact every two weeks. The age of the children was negatively related to the frequency of contact, *r* = −0.30, *p* = 0.038. Parents with older children had less frequent contact than those with younger children. Again, the frequency of contact with the EI provider was negatively related to the amount of working hours for fathers, *r* = −0.49, *p* = 0.014, but not mothers, *r* = −0.08, *p* = 0.728.

In addition to in-home EI sessions, parents reported their attendance at courses/trainings offered at the EI center. Fathers and mothers attended with the same frequency, *U* = 0.276, *p* = 0.704 (Fathers, M = 2.54, SD = 1.01; Mothers, M = 2.63, SD = 0.101; range 1–4). Among fathers, 37.5% never (12.5%) or rarely (25.0%) attended, whereas the majority of fathers did attend courses/trainings held at EI center—62.5% attended often (45.8%) or always (16.7%). Among mothers, 37.5% never (16.7%) or rarely (20.8%) attended courses at the EI center, while 62.5% of mothers attended often (41.7%) or always (20.8%). Neither fathers’ nor mothers’ number of hours of employment were related to EI center course/training attendance. Neither fathers’ educational level nor child age was related to attendance, although mothers’ educational level was, *r* = 0.42, *p* = 0.043. Mothers with higher levels of education more frequently attended courses/trainings.

### 3.5. Perceived Support by EI Provider

Fathers as well as mothers reported a high sense of support by their EI provider and did not differ in their perceived support (Table 3). At the item level (Table 5), both fathers and mothers perceived less support from their EI provider in one important area—handling reactions from family, friends, and society regarding the child’s hearing loss. Child age and parental educational level did not affect these results.

### 3.6. Associations of Parental Self-Efficacy and Involvement

Significant associations were found between PSE and involvement-related variables for fathers (see Table 6). Fathers who rated themselves higher in PSE also reported higher levels of involvement in their child’s intervention, perceived greater support from their EI Provider, and attended more in-home EI sessions. In addition, fathers who reported being more involved in their child’s intervention also perceived greater support by the EI Provider and attended more EI sessions in the home. Similar associations were found for mothers of children who are DHH. The data revealed only a negative significant association between mothers’ feelings of involvement and the impact of their child’s hearing status on their parenting. Mothers who experienced more challenges were less involved in EI.

## 4. Discussion

Although FCEI for children who are DHH aims to support caregivers in promoting their child’s development by addressing the needs of the whole family, the focus of research with this population was centered, to a large extent, on mothers [13]. Less is known about the experiences of fathers, such as the facilitators and barriers to promoting their involvement in EI and their beliefs about their PSE in their role as fathers. The present study aims to provide insights into the perspectives and needs of both fathers and mothers of young children who are DHH and receive FCEI services.

### 4.1. Similarities among Fathers and Mothers

FCEI for children who are DHH is intended to promote parental self-efficacy (PSE). In other studies involving families of children with disabilities, higher levels of PSE have been linked to better child outcomes, improved parent–child interaction, and enhanced parental wellbeing [17,56]. In the current study, both fathers and mothers reported high levels of PSE. This result is promising, as it indicates that, in general, fathers and mothers believe that they are able to successfully raise their child who is DHH.

Both fathers and mothers felt supported by their primary EI provider in promoting their child’s language and social–emotional development; indeed, these are areas in which FCEI has been shown to be effective in a variety of studies [1,44,57,58]. Both fathers and mothers indicated that they felt supported by their primary EI provider and gained knowledge that assisted in dealing with their child being DHH in daily situations. However, they felt less supported in handling their child’s hearing assistive devices (where applicable) and in dealing with reactions from family and friends.

### 4.2. Differences among Fathers and Mothers

In the present study, parents were asked about the frequency of participation in EI and their perceived support from their primary EI provider. Mothers were more frequently present for in-home EI sessions than fathers; however, no differences were found in their attendance at courses offered at the EI center. The courses taught at the EI center (focusing on such topics as acquisition of sign language, facilitating spoken language abilities, and fostering social–emotional development) are often offered in the evening, whereas the in-homes sessions are nearly always scheduled during the day. Fathers who worked more hours per week attended the in-home sessions less frequently, although the amount of time spent at work did not seem to negatively impact their attendance at courses taught at the EI center. The same association was not found in mothers. This may imply that fathers’ availability is less considered when scheduling in-home EI sessions. In a study conducted in Germany [30], around 75% of fathers indicated that they were not included when FCEI appointments were scheduled. Actively addressing fathers’ needs by considering the barriers to their involvement, such as offering more flexibility in scheduling in-home EI sessions and/or using telepractice technology, might make it to more feasible for fathers to participate. (A study of FCEI services provided in-person versus via telepractice did not find that delivery mode impacted caregivers’ PSE or involvement, suggesting that both delivery modes could be effective for delivery of FCEI services [28].) This seems important to consider, given that the fathers in our study who more frequently joined the in-home sessions reported higher levels of PSE and were more involved in EI.

The frequency of joining the in-home sessions was not related to PSE in mothers. This finding is in line with a recent study of Davenport and colleagues [26] who investigated the effects of EI dosage on PSE among parents of children who are DHH. They found no differences in PSE between parents who participated frequently or less frequently in EI services. The majority of participants were mothers (94%). The findings of the current study replicate this finding for mothers, but not for fathers. This again points to potential differences among fathers and mothers related to their FCEI needs.

Importantly, high levels of PSE were found in both fathers and mothers. However, mothers reported higher levels of PSE in the following domains: emotions and affection; play and enjoyment; empathy and understanding; control; and learning and knowledge. This finding may allude to a need to foster more fun, social exchanges between fathers and their child who is DHH. For example, EI providers may wish to include in their repertoire of “tools” the promotion of play and shared activities, exploration and demonstration of emotion, and strategies for building empathy and understanding of others. While strategies that build upon these skills can be helpful for all families, they may be especially beneficial for EI sessions involving fathers.

Fathers who reported higher levels of PSE also reported that they were highly involved in the intervention, often attended the in-home EI sessions, and perceived much support from EI providers. Those fathers who worked fewer hours per week felt more confident in raising their child who is DHH. These associations were not found in mothers. The current finding that PSE and involvement in EI are related in fathers, but not in mothers, was also reported by Zaidman-Zait and colleagues [48]. They argue that the lack of this association in mothers might be explained by the fact that mothers, as part of their role identity, feel obligated to attend and participate in their child’s EI regardless of their PSE levels. While neither of these studies can attribute causality, their findings do imply potentially different patterns of self-efficacy and involvement between mothers and fathers.

Family involvement is a foundational principle in FCEI and has a number of implications (including being associated with better language outcomes) [46]. Mothers endorsed greater involvement when their child was older, whereas fathers did not. Additionally, both fathers and mothers of older children had less frequent contact with their primary EI provider than those of younger children (it is important to note that the oldest children in the current sample were four years of age, the age at which most children in the Netherlands “age out” of FCEI). As children grow, the mothers’ involvement seems to increase, despite having less frequent contact with the EI provider. Perhaps this is related to the EI providers’ success in promoting involvement and providing mothers with knowledge and skills, rendering less frequent contact sufficient. Parents were queried about the frequency of contact with their primary EI provider; however, this frequency rating did not reflect parents’ contact with other EI providers. Had parents been asked about all of their EI providers, this may have yielded different results regarding frequency of contact.

For fathers, the relationship between their involvement in EI and the amount of perceived support from their EI provider is particularly strong. This association was also reported by Hintermair and Sarimski [30]. A greater understanding of the factors that facilitate or hinder involvement for both fathers and mothers can help EI providers to individualize their support and increase involvement by parents of both genders.

### 4.3. Limitations

In the current study, parents indicated the frequency of their contact with their primary EI provider. Half of the parents reported having contact < 1 time per month. Additionally, parents of older children had less frequent contact than those with younger children. However, inquiring about the frequency of contact with the primary EI provider may not have sufficiently captured the extent to which families are involved with EI services. In addition to the primary EI provider, additional specialists can be involved, depending on the needs, strengths, and values of the child and family. These professionals can include, for example, sign language teachers, speech-and-language therapists, and teachers of the deaf. Additionally, beginning at the age of 18 months, children in the Netherlands can also participate in specialized EI groups for DHH children at the EI center for a portion of the day, twice per week. Children who are DHH and participate in EI-center-based groups, along with their parent(s), have more contact with other EI providers and specialists.

Another limitation of the current study is the relatively small sample size. It is recommended that future studies examining FCEI processes and outcomes include both fathers and mothers of children enrolled in FCEI to examine whether the findings of the present study can be replicated, and to further enhance the understanding about the differential strengths and needs of fathers and mothers of children who are DHH enrolled in FCEI.

### 4.4. Implications for FCEI

FCEI programs strive to support families in the promotion of the development of their child who is DHH. Parents who participated in the current study reported feeling highly supported by their primary EI provider. However, both fathers and mothers reported that they felt less supported in managing their child’s hearing technology-related needs and addressing questions and reactions from others, including family and friends. This suggests a potential area that EI providers may wish to address directly with caregivers.

Drawing inferences from the responses of fathers and mothers in the current study, the following “implications for FCEI” are offered for consideration:Ask fathers and mothers what “involvement in FCEI services” means to them;Ask fathers their preferences for how they wish to be involved and engaged with supporting their child who is DHH;Ask fathers to identify any known barriers and facilitators to their involvement;Once barriers and facilitators are identified, work collaboratively to see whether the barriers can be minimized and the facilitators enhanced to allow a greater involvement and engagement with FCEI;Consider conducting a focus group to better understand fathers’ experiences with FCEI;Include all caregivers/parents in correspondence with the EI provider and the EI program. For example, include spaces for more than one email contact and reach out to all caregivers (including mothers and fathers, as well as others who might be in a caregiving role) when scheduling in-home EI sessions or exchanging information with the family;Offer flexible scheduling of in-home EI sessions. Consider scheduling sessions at a time when all caregivers can be present;Use telepractice approaches to increase engagement of all caregivers when possible or feasible;Consider incorporating strategies and tools to assist caregivers in promoting play and social–emotional development in their children who are DHH. (For example, in the Netherlands, a course for parents about how to promote social–emotional development and theory of mind among children who are DHH has been developed);Take care to not assume that fathers do not want to be involved in FCEI; challenge one’s own assumptions that “mothers are more involved” and—instead—consider how to make FCEI more inclusive of fathers.

## Figures and Tables

**Figure 1 jcm-11-00492-f001:**
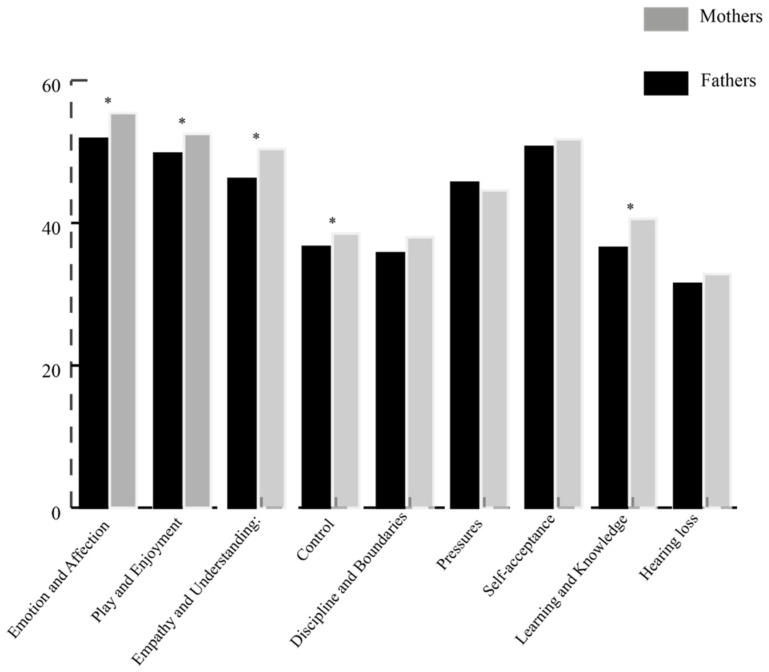
Mean scores of fathers’ and mothers’ parental self-efficacy subscales. * *p* < 0.05.

**Table 1 jcm-11-00492-t001:** Demographic characteristics of participants in the present study.

		Fathers	Mothers
Parents’ characteristics			
Mean age in years (SD)		34.96 (4.33)	32.97 (3.72)
Age range in years		28–43	25–39
Level of education			
	Elementary school education	1	1
	Vocational/High school education	6	7
	University/college education	17	16
Hearing status			
	Typical hearing	21	23
	Hearing loss	3	1
Child characteristics			
Mean age (in months)		27.38 (12.92)	
Age range (in months)		9–48	
Gender (%)			
	Male	14 (58%)	
	Female	10 (42%)	
Degree of hearing loss			
	40–60 dB	12 (50%)	
	60–90 dB	5 (21%)	
	>90 dB	7 (29%)	
Amplification			
	Hearing aids	13 (54%)	
	Cochlear implants	7 (29%)	
	BaHa	3 (13%)	
	ABI	1 (4%)	
Additional challenges		8 (33%)	

**Table 2 jcm-11-00492-t002:** Psychometrics of the questionnaires.

	No. of Items	Range	Cronbach’s Alpha
Parental self-efficacy subscales (TOPSE)			
Emotion and Affection	6	0–10	0.60
Play and Enjoyment	6	0–10	0.79
Empathy and Understanding	6	0–10	0.86
Control	5	0–10	0.54
Discipline and Boundaries	6	0–10	0.73
Pressures	5	0–10	0.76
Self-acceptance	5	0–10	0.82
Learning and Knowledge	5	0–10	0.65
Hearing loss	4	0–10	0.78
Fathers of Children with Developmental Challenges—Revised to include mothers			
Impact on parenting	12	1–5	0.79
Involvement in FCEI	5	1–5	0.78
Perceived support by FCEI	6	1–5	0.85

**Table 3 jcm-11-00492-t003:** Means and standard deviations of fathers’ and mothers’ study variables (parental self-efficacy, impact on parenting, involvement in intervention, and perceived support).

	Fathers	Mothers	*U*	*p*
	*n* = 24	*n* = 24		
	Mean (SD)	Mean (SD)		
Parental self-efficacy total	406.08 (42.13)	432.88 (37.87)	198.00	0.063
Impact on parenting	1.83 (0.42)	1.82 (0.42)	287.50	0.992
Involvement in intervention	4.06 (0.55)	4.31 (0.69)	194.00	0.050
Perceived support	3.85 (0.56)	4.00 (0.63)	240.00	0.319

**Table 4 jcm-11-00492-t004:** Impact of child’s deafness on parents’ experience.

Fathers of Children with Developmental Challenges—Revised to Include Mothers—Subscale “Impact on Parenting”	% Strongly Not Agree/Not Agree (1/2)		% Not Sure (3)		% Agree/Strongly Agree (4/5)	
	F	M	F	M	F	M
I do not have the energy to be able to help with my child’s therapies.	87.5	95.8	12.5	4.2	0	0
His/her hearing loss make me want to avoid caring for my child.	95.8	100	4.4	0	0	0
My spouse/partner does not believe I can handle the demands of my child’s hearing loss.	100	100	0	0	0	0
I cannot handle the difficulties that come with my child’s hearing loss. *	87.5	95.8	12.5	4.2	0	0
Many of my ideas about parenthood have changed because I have a DHH child.	91.7	79.2	4.2	8.3	4.2	12.5
I am not sure I am able to obtain whatever information I need about her/ his hearing loss. *	87.5	95.8	12.5	4.2	0	0
His/her hearing loss gets in the way of our relationship.	95.8	100	4.2	0	0	0
I find myself thinking that the dreams I had for my child will probably not happen.	83.3	79.2	8.3	12.5	8.3	8.3
Being involved in his/her therapies is a lot for me to handle.	58.3	54.2	29.2	29.2	12.5	16.7
Having a DHH child is more difficult than I expected.	79.2	70.8	16.7	20.8	4.2	8.3
Having a DHH child has a large impact on the quality of time that we spend together.	83.3	91.7	16.7	4.2	0	4.2
I do dwell on my child’s diagnosis. *	33.3	16.7	12.5	25	54.2	58.3

*** = reversed. Abbreviations F = Fathers, M = Mothers.

**Table 5 jcm-11-00492-t005:** Perceived support by FCEI.

My Early Interventionist Supports Me in …	% Strongly Not Agree/Not Agree (1/2)		% Not Sure (3)		% Agree/Strongly Agree (4/5)	
	F	M	F	M	F	M
Promoting my child’s language development.	0	0	16.7	12.5	83.3	87.5
Promoting my child’s social–emotional development.	0	0	29.2	25.0	70.8	75.0
Coping with my child’s diagnosis.	8.3	8.3	25.0	20.8	66.7	70.8
Handling reactions from my family, friends and society.	12.5	12.5	37.5	25.0	50.0	62.5
In managing and care for my child’s hearing device.	4.2	0	33.3	29.2	62.5	66.7
Coping with the HL of my child in daily situations.	0	0	12.5	16.7	87.5	83.3

**Table 6 jcm-11-00492-t006:** Spearman correlations between parental self-efficacy, impact on parenting, involvement in intervention, and perceived support.

	1	2	3	4	5	6
(1) Parental self-efficacy						
Fathers						
Mothers						
(2) Impact of hearing loss on parenting						
Fathers	−0.14	-	-	-	-	-
Mothers	−0.36	-	-	-	-	-
(3) Parental involvement in child intervention						
Fathers	0.57 **	−0.37	-	-	-	-
Mothers	0.31	−0.43 *	-	-	-	-
(4) Perceived support						
Fathers	0.44 *	−0.32	0.72 **	-	-	-
Mothers	−0.04	0.39	0.22	-	-	-
(5) Frequency of house visit attendance						
Fathers	0.41 *	0.10	0.46 *	0.18	-	-
Mothers	0.09	−0.07	0.27	0.17	-	-
(6) Frequency of center attendance						
Fathers	0.31	−0.10	0.38	0.23	−0.10	-
Mothers	−0.10	0.06	0.14	0.37	−0.26	-

* *p* < 0.05, ** *p* < 0.01.

## Data Availability

The data presented in this study are available upon request from the corresponding author. The data are not publicly available because consent for this was not obtained from the participants.

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
