# Peer review of "Family-Centered Early Intervention (FCEI) Involving Fathers and Mothers of Children Who Are Deaf or Hard of Hearing: Parental Involvement and Self-Efficacy"

_jcm, 2022, doi:10.3390/jcm11030492_

Round 1

Reviewer 1 Report

A well done paper addressing an important issue about mothers v fathers.  Two minor concerns are:

  1. In table 2, three subscales were examined due to low reliability and an item was dropped.  Since Emotion and Affection has a low Alpha, why was it not examined?

2. Table 4 and Table 5 are based on items where the subscale or scale showed no significant differences.  Since there are no differences that are significant, I would recommend not examining differences on the items.  However, the results and follow up discussion seem more addressed on the descriptive statistics for the items without the distinction between mother and father.  Therefore, I would recommend the two tables to be reduced to descriptive statistics for the items with mothers and fathers combined.  

Author Response

Please see the attachment. Thank you for your comments. 

Reviewer 2 Report

Please see page 6 for comments

Author Response

(The authors gave the same response as above.)
